# Prevalence and Risk Factors of Abnormal Glucose Metabolism and New-Onset Diabetes Mellitus after Kidney Transplantation: A Single-Center Retrospective Observational Cohort Study

**DOI:** 10.3390/medicina58111608

**Published:** 2022-11-07

**Authors:** Carlo Alfieri, Evaldo Favi, Edoardo Campioli, Elisa Cicero, Paolo Molinari, Mariarosaria Campise, Maria Teresa Gandolfo, Anna Regalia, Donata Cresseri, Piergiorgio Messa, Giuseppe Castellano

**Affiliations:** 1Nephrology, Dialysis and Transplantation, Fondazione IRCCS Ca’ Granda Ospedale Maggiore Policlinico, 20122 Milan, Italy; 2Department of Clinical Sciences and Community Health, Università degli Studi di Milano, 20122 Milan, Italy; 3Kidney Transplantation, Fondazione IRCCS Ca’ Granda Ospedale Maggiore Policlinico, 20122 Milan, Italy; 4Nephrology, IRCCS Ospedale Policlinico San Martino, 16132 Genova, Italy

**Keywords:** diabetes mellitus, glucose metabolism, kidney transplantation, risk factors, immunosuppression

## Abstract

*Background and objectives:* New-onset diabetes after transplantation (NODAT) represents a primary cause of morbidity and allograft loss. We assessed prevalence and risk factors for NODAT in a population of Italian kidney transplant (KT) recipients. *Methods:* Data from 522 KT performed between January 2004 and December 2014 were analyzed. Participants underwent clinical examination; blood and urine laboratory tests were obtained at baseline, one, six, and 12-month of follow-up to detect glucose homeostasis abnormalities and associated metabolic disorders. An oral glucose tolerance test (OGTT) was performed at six months in 303 subjects. *Results:* Most patients were Caucasian (82.4%) with a mean age of 48 ± 12 years. The prevalence of abnormal glucose metabolism (AGM) and NODAT was 12.6% and 10.7%, respectively. Comparing characteristics of patients with normal glucose metabolism (NGM) to those with NODAT, we found a significant difference in living donation (16.6% vs. 6.1%; *p* = 0.03) and age at transplant (46 ± 12 vs. 56 ± 9 years; *p* = 0.0001). Also, we observed that patients developing NODAT had received higher cumulative steroid doses (1-month: 1165 ± 593 mg vs. 904 ± 427 mg; *p* = 0.002; 6-month:2194 ± 1159 mg vs. 1940 ± 744 mg; *p* = 0.002). The NODAT group showed inferior allograft function compared to patients with NGM (1-year eGFR: 50.1 ± 16.5 vs. 57 ± 20 mL/min/1.73 m^2^; *p* = 0.02). NODAT patients were more likely to exhibit elevated systolic blood pressure and higher total cholesterol and triglyceride levels than controls. *Conclusions*: The prevalence of NODAT in our cohort was relatively high. Patient age and early post-transplant events such as steroid abuse are associated with NODAT development.

## 1. Introduction

Kidney transplantation (KT) provides a survival advantage and superior quality of life over dialysis [1,2]. However, due to their complex comorbidity and the long-term effects of chronic immunosuppression, transplant recipients remain at increased risk of death than the general population [3]. In particular, cardiovascular complications represent the leading cause of post-transplant mortality and a primary determinant of premature allograft loss [4,5,6]. Among the many potential contributing factors involved in the pathogenesis and progression of post-transplant cardiovascular disease (CVD) [7,8] new-onset diabetes mellitus after transplantation (NODAT) certainly plays a role as it is frequently associated with infections, metabolic disorders, abnormal proteinuria, and major cardiovascular events (MACE). Progression to overt diabetic nephropathy may also occur, with devastating consequences for the transplanted kidney [9]. 

Even though the International Congress Guidelines [10] and the American Diabetes Association (ADA) [11] state that NODAT should be defined using the same diagnostic criteria as for type 2 diabetes mellitus in the non-transplant population, NODAT is a specific form of type 2 diabetes characterized by impaired β-cell insulin secretion acting on a background of insulin resistance [9]. Importantly, when diagnosing NODAT, acute infections and other forms of post-transplant hyperglycemia such as stress-induced or immunosuppression-induced hyperglycemia should be ruled out [10]. 

The prevalence of glucose regulation abnormalities in KT recipients is extremely variable and mostly depends on the diagnostic criteria employed [12]. Data on NODAT are heterogeneous as well, suggesting a peak incidence within three months of transplant and an overall occurrence between 4% and 27% [13]. This relevant variability probably reflects differences in study populations (namely, demographics, lifestyle, transplant characteristics, and immunosuppression) and the continuous evolution of the definition of the disease over time. 

To date, several modifiable and non-modifiable risk factors for NODAT have been identified [9]. Putative post-transplant modifiable risk factors are overweight [14], vitamin D deficiency [15], hypomagnesemia [16], cytomegalovirus (CMV) infection [17], sedentary behavior [18], and immunosuppressive medications [19]. Non-modifiable risk factors include familial predisposition, age, and ethnicity. 

Managing KT patients with NODAT is often challenging as they frequently show fluctuating renal function and increased susceptibility to drug-related adverse events [9]. Possible interactions between oral hypoglycemic agents and immunosuppressants must be carefully evaluated, and the dose and type of insulin should be determined preferring an individualized approach [9]. Undoubtedly, prophylactic strategies would represent the best option, but potential interventions are limited by the lack of standardized screening protocols and by the scarcity of evidence supporting a specific approach over the other ones. Both patients on the transplant waiting list and transplant recipients should be assessed periodically for metabolic abnormalities, and recognized risk factors for NODAT should be promptly addressed [12]. Preventive measures may comprise aggressive body weight control, caloric intake restriction, implementation of regular exercise activity, and tailored immunosuppression such as calcineurin inhibitor (CNI) minimization or steroid sparing schemes [9]. 

We aimed to evaluate the prevalence of glucose metabolism abnormalities in a cohort of Italian contemporary KT recipients, focusing on the early post-transplant course and investigating post-transplant modifiable risk factors.

## 2. Materials and Methods

In this single-center retrospective observational cohort study with a prespecified follow-up of 12 months, we analyzed data from patients who had undergone KT at the Fondazione IRCCS Ca’ Granda Ospedale Maggiore Policlinico (Milan, Italy) between January 2004 and December 2014. Patients with pre-transplant diabetes, multiple organ transplantation, and re-transplantation were considered in the cohort studied. Exclusion criteria were recipient age <18 years and allograft loss within six months of surgery. 

Treatments and procedures herein reported were in accordance with the ethical standards of the institutional committee at which it was conducted (Fondazione IRCCS Ca’ Granda Ospedale Maggiore Policlinico Ethical Committee, Protocol ID 4759-1837/19), as well as with the 1964 Helsinki declaration and its later amendments, or comparable ethical standards. All participants consented for research purposes at the time of activation on the transplant waiting list and their willingness to participate in future non-interventional research projects was confirmed before the transplant procedure. Donor data, organ details, recipient characteristics, and study-related outcomes were recorded in a central database by dedicated staff (as per local practice) and reviewed by the authors in December 2021. 

Enrolled subjects were thoroughly assessed at the time of transplant, and serial clinical and laboratory evaluations were performed at one (T1), six (T6), and 12 (T12) months of follow-up. Analyzed data included: donor type, cold ischemia time (CIT), recipient familiarity for diabetes mellitus, past medical history, age, gender, body mass index (BMI), primary renal disease (PRD), renal replacement therapy (RRT), dialysis vintage, hepatitis B virus (HBV), hepatitis C virus (HCV) and CMV serostatus, immunosuppressive therapy, renal function, systolic (SBP) and diastolic (DBP) blood pressure, glucose-homeostasis-related parameters, hemoglobin, serum uric acid, intact parathyroid hormone (iPTH), 25-hydroxivitamin D, calcium, phosphorus, and albumin levels, lipid profile, and alkaline phosphatase (AP) concentration. All samples were collected and processed centrally, after 12 h of fasting. 

Glucose-homeostasis-related parameters included fasting plasma glucose (FPG), basal insulinemia, glycated hemoglobin (HbA1c) levels, and homeostasis model assessment (HOMA) index [20] at T1, T6, and T12. Compliant patients without pre-transplant history of diabetes or overt NODAT were also evaluated with oral glucose tolerance test (OGTT) at T6. The HOMA index is a method used to quantify insulin resistance and β-cell function from basal blood glucose and insulin concentrations, using the following formula: fasting blood glucose (mg/dL) × fasting blood insulin (mIU/L)/405. Subjects without insulin resistance have a HOMA index between 0.23 and 2.5. OGTT was carried out early in the morning, after a minimal fasting period of 12 h. Patients were administered an oral glucose load (75 g) and a venous blood sample was taken at baseline, 30, 60, and 120 min to assess glucose and insulin levels. According to FPG and/or OGGT, patients were sorted into four categories: (1) Normal Glucose Metabolism (NGM), fasting glycemia < 110 mg/dL or 120 min glycemia < 120 mg/dL; (2) Impaired Fasting Glucose (IFG), fasting glycemia between 110 and 125 mg/dL; (3) Impaired Glucose Tolerance (IGT), 120 min glycemia between 140 and 199 mg/dL; and (4) NODAT, fasting glycemia >125 mg/dL or 120 min glycemia > 200 mg/dL. For analysis purposes, recipients with IFG or IGT were merged into a larger group named Abnormal Glucose Metabolism (AGM). Similarly, patients with pre-transplant diabetes (Pre-KT-DM) or post-transplant diabetes (NODAT) were grouped together as Overall Diabetes Mellitus (ODM). 

Renal function was evaluated by serum creatinine concentration (SCr), estimated glomerular filtration rate (eGFR) according to the Modification of Diet in Renal Disease (MDRD) formula [21], and 24 h proteinuria (immune-turbidimetric method).

For iPTH detection and quantification, we used the electro-chemiluminescence immune-assay (ECLIA) and the E170 module for the Modular Analytics (Roche Diagnostics, Basel, Switzerland). The measurement range was 1.20–5000 pg/mL whereas the conversion was pg/mL × 0.106 = pmol/L (normal range, 15–65 pg/mL).

Serum vitamin D levels were assessed using an enzyme immune-assay (Kit EIA AC-57 FI, Immuno-Diagnostic System, Boldon, UK) with highly specific 25-hydroxivitamin D sheep antibody and enzyme-labeled avidin (horseradish peroxidase). The sensitivity threshold was 5 nmol/mL (2 ng/mL). The specificity of the antiserum was tested with the following analytes calibrated at the level of 50% binding of the zero standard: 25-hydroxivitamin D3 (cross-reactivity, 100%), 25-hydroxivitamin D2 (cross-reactivity, 75%), 24,25-dihydroxyvitamin D3 (cross-reactivity, 100%), cholecalciferol (cross-reactivity, <0.01%), and ergocalciferol (cross-reactivity, >0.30%). The intra-assay precision was calculated from 10 duplicate determinations of two samples each, performed in a single assay (CV between 5.3% and 6.7%). Inter-assay precision was calculated from duplicate two-sample determinations performed in 11 assays (CV between 4.6% and 8.7%). 

Categorical variables were described using proportions and percentages. Continuous variables were reported as mean (± standard deviation) or median (25°–75° percentiles) in case of abnormal distributions. Data were compared using Fisher’s exact test, chi-square test, Student’s t-test, Wilcoxon–Mann–Whitney U test, or ANOVA as appropriate. We assessed the predictive ability of a pool of variables for the risk of NODAT building linear and logistic regression models for univariate and multivariate analysis. Significance was defined as *p* value <0.05. Analyses were performed using Statistica 10 (StatSoft, Palo Alto, CA, USA) and SPSS 20 (IBM, Armonk, NY, USA).

## 3. Results

Between January 2004 and December 2014, 531 adult patients received a KT at our institution. Nine (9/531, 1.69%) of them experienced allograft loss within six months of transplant and were excluded from the study. The final analysis was carried out on 522 participants. 

The study population mostly included Caucasian recipients (430/522, 82.4%) on RRT (482/522, 92.3%), with a small disproportion between males (n = 295) and females (n = 227). The mean age at transplant was 48 ± 12 years and the vast majority of patients (444/522, 85.1%) received a deceased donor kidney (Table 1). 

At six months of follow-up, 303 (303/522, 58.05%) participants underwent OGTT. As specified above, non-compliant patients (166/522, 31.8%) as well as those with pre-transplant history of diabetes mellitus (14/522, 2.7%) or NODAT diagnosis before T6 (34/522, 6.5%) were exempt from the test. 

In our cohort, 73.9% of patients maintained an NGM. During the observation period, the proportion of recipients developing AGM or NODAT was 10.7% and 12.6%, respectively. Among the latter group, 51.5% were diagnosed NODAT within six months of transplant, whereas in 24.2% of patients, NODAT was diagnosed by OGTT. In other 24.2% of patients, NODAT was diagnosed after OGTT. Comparing the baseline characteristics of the patients in the NGM group with those of the recipients with AGM or NODAT, we observed a significant difference in terms of familial predisposition to diabetes (NGM: 66/386, 17.1% vs. NODAT: 23/66, 34.8%; *p* = 0.0008), previous exposure to HBV infection (NGM: 8/386, 2.1% vs. AGM: 6/56, 10.7%; *p* = 0.0005), living donation (NGM: 64/386, 16.6% vs. NODAT: 4/66, 6.1%; *p* = 0.0271), and age at transplant (NGM: 46 ± 12 years vs. AGM: 51 ± 11 years vs. NODAT: 56 ± 9 years; *p* = 0.01 and *p* = 0.0001). Patients who eventually developed AGM were also more likely to have autosomal dominant polycystic kidney disease (NGM: 72/386, 18.6% vs. AGM: 18/56, 32.1%; *p* = 0.0191) or vascular nephropathy (NGM: 95/386, 24.6% v AGM: 21/56, 37.5%, *p* = 0.0404) as the primary cause of ESRD. On the contrary, ethnicity, gender, RRT modality, dialysis vintage, and previous exposure to HCV or CMV infections were substantially similar (Table 1 and Table 2).

Analyzing the exposure to immunosuppressive medications, we found that patients with NODAT had received higher cumulative steroid doses than those with NGM, particularly at one (1165 ± 593 mg vs. 904 ± 427 mg; *p* = 0.002) and six (2194 ± 1159 mg vs. 1940 ± 744 mg; *p* = 0.002) months of follow-up. Regarding other immunosuppressants, our data did not show any significant difference. However, the preferential use of tacrolimus (458/522, 87.7%) over cyclosporine, and the limited number of recipients on sirolimus or everolimus (15/522, 2.9%) did not allow a proper comparison between groups. 

As detailed in Table 3, several clinical and laboratory parameters were evaluated during the follow-up. 

Notably, recipients with NODAT showed higher SCr concentration and lower eGFR than patients with NGM, at every time point of the study (1-month eGFR: 50.5 ± 23.4 vs. 57 ± 21 mL/min/1.73 m^2^, *p* = 0.03; 6-month eGFR: 46.7 ± 15.3 vs. 56 ± 19 mL/min/1.73 m^2^, *p* = 0.001; 1-year eGFR: 50.1 ± 16.5 vs. 57 ± 20 mL/min/1.73 m^2^, *p* = 0.02). Furthermore, the NODAT group was characterized by higher SBP (6-month: 140 ± 25 vs. 131 ± 19 mmHg, *p* = 0.001; 1-year: 140 ± 21 vs. 130 ± 17 mmHg, *p* = 0.001), BMI (1-month: 25 ± 4 vs. 23 ± 4 kg/m^2^, *p* = 0.0001; 1-year: 26 ± 4 vs. 24 ± 4 kg/m^2^, *p* = 0.006), and uric acid levels (6-month: 7.1 ± 2 vs. 6.5 ± 1.5 mg/dL, *p* = 0.008; 1-year: 7.3 ± 2 vs. 6.5 ± 1.5 mg/dL, *p* = 0.001) than the NGM one. In addition to worse glucose-homeostasis-related parameters such as FPG, HbA1c, and HOMA index (1-month: 2.4, 1.5–3.6 vs. 2.5, 1.5–3.7, *p* = 0.001; 6-month: 2.1, 1.4–3.1 vs. 1.5, 1.1–2.2, *p* = 0.001; 1-year: 1.9, 1.2–3 vs. 1.6, 1.1–2.1, *p* = 0.003), NODAT patients exhibited higher total cholesterol (1-year: 211 ± 51 vs. 199 ± 46 mg/dL; *p* = 0.05) and triglyceride levels (1-year: 173 ± 84 vs. 146 ± 67 mg/dL; *p* = 0.01) with reduced HDL cholesterol concentrations (1-year: 50 ± 14 vs. 58 ± 19 mg/dL; *p* = 0.003) compared to recipients with NGM. Mineral metabolism biomarkers including iPTH, 25-hydroxivitamin D, calcium, phosphorus, and AP levels, as well as serum albumin and 24 h proteinuria were not significantly different.

According to our multivariate analysis (Table 4), recipient age at transplant, BMI, renal function at one month of follow-up, and cumulative steroid dose within 30 days of transplant were associated with NODAT development. 

## 4. Discussion

We performed a retrospective observational study to assess the prevalence of glucose metabolism derangements in a population of contemporary KT recipients in Italy. Possible relationships between donor-, recipient-, or transplant-related factors and NODAT development during the first post-transplant year were also investigated.

Diabetes mellitus represents the leading cause of ESRD worldwide [22]. According to recent data, up to 30% of the diabetic patients in the USA, UK, Western Europe, or Japan show signs of chronic kidney disease (CKD) [23,24,25,26], with unsustainable costs for both private and public health care systems. In our series, the proportion of KT candidates with diabetes mellitus was 2.7%, much lower than the prevalence reported at national level in the general population (6.4%) and inferior to those recorded in countries with similar social, economic, or behavioral characteristics such as Spain (10.3%), France (5.3%), Portugal (9.1%), or Greece (6.4%) [27,28]. The reason behind this difference is difficult to determine and may recognize several contributing factors including the small sample size of the cohort, the preponderance of Caucasian participants (82.4%) [29,30], the young age at transplant (mean, 48 years), the relatively high educational and financial background of the population living in the Lombardia region [31,32], the discrepancy in the quality of the public health care providers across the country [33], the lack of a pancreas or pancreatic islet transplantation program at our institution, and, perhaps, the missed opportunity to detect further diabetic subjects with a systematic use of OGTT at the time of enlistment [12]. 

Previous studies have shown that in potential KT recipients, abnormal glucose homeostasis is more frequent than the general population [34], probably reflecting the increase in basal insulin resistance associated with CKD [35]. The number of patients developing IFG, IGT, or NODAT within the first post-transplant year in the present study (overall, >23%) somehow confirms the increased susceptibility of ESRD patients to diabetes [28] and, once again, reflects the sub-optimal application of diabetes screening protocols during the pre-transplant phase [36]. 

As for pre-transplant diabetes, the prevalence of AGM (10.7%) or NODAT (12.6%) among our KT recipients was lower than that reported in the literature (> 20%) [12,13,37]. Such reassuring results may be partially due to the favorable baseline characteristics of the patients enrolled [38], mostly Caucasian [39,40], young [9,40], non-obese (mean BMI, 23) [40], and without a history of chronic HBV (3.3%) or HCV (6.7%) [41] infection. However, other variables may have played a role, such as the preferred use of cyclosporine over tacrolimus in recipients at increased risk of NODAT [42,43], routine application of CNI-minimization protocols [44], paucity of patients receiving mTORi (<3%) [45], relatively low doses of steroid administered at induction (≤750 mg) [46], frequent (twice weekly) CMV viremia assessment during the early post-transplant phase [17], dedicated counselling for optimal weight control after transplant [47], and aggressive outpatient monitoring for hypertension [48], lipid disorders [44], magnesium [16], or vitamin D deficiency [9]. Unfortunately, the well-established immunosuppressive protocols based mostly on steroids, tacrolimus, and MMF made the analysis of the impact of therapy on NODAT development impossible. Prompt detection and treatment of recipients with IFG or IGT is also key factor for the prevention of NODAT [49]. Accordingly, since 2010, our institution has been providing KT patients continuous evaluation of glucose metabolism-related parameters such as FPG, HbA1c, and basal insulinemia. All recipients have also been encouraged to undergo OGTT within six months of transplant, and we are now working on pre-transplant screening implementation strategies. 

The importance of pre- and post-transplant OGTT for early detection of glucose metabolism abnormalities cannot be emphasized enough [36]. The fact that about half of the NODAT cases recorded in our series have occurred in the first five months of follow-up highlights the importance of pre-transplant predisposing factors and early post-transplant events, confirming the need for standardized dynamic glucose testing throughout the entire transplant process [12]. It is worth mentioning that, despite sub-optimal adherence to the screening protocol, the OGTT performed six months after transplant was able to identify a significant number of patients without clinically evident NODAT (25% of the total) [12].

In line with other studies, the present analysis demonstrates the association between specific demographic and clinical characteristics at the time of transplant and future NODAT development. In particular, familial predisposition to diabetes [9], older age [50], and higher BMI [51] have been consistently reported as important contributing factors. On the contrary, less clear remain the possible relationships between NODAT and autosomal polycystic kidney disease [52], HCV [53], or HBV infection [41]. 

Donor and transplant characteristics were also evaluated. Basically, our findings suggest that deceased donor recipients are more likely to develop NODAT than their living donor counterpart and confirm the association between high-dose steroid use and post-transplant glucose metabolism abnormalities. The beneficial effect of living donation on NODAT susceptibility recognizes several reasons. First of all, living donor kidneys ensure better long-term allograft survival and function than deceased donor ones [54,55]. Secondly, due to better donor–recipient matching and lower incidence of delayed graft function, living donor transplants require milder immunosuppression. Lastly, living donation is associated with higher rates of pre-emptive transplantation with superior recipient- and allograft-related outcomes compared to dialysis vintage patients [56]. The observation that higher steroid doses during the early post-transplant phase increase the risk of NODAT development is certainly relevant, but not anyhow surprising, as several studies have demonstrated that rapid steroid withdrawal and steroid minimization immunosuppressive strategies can greatly reduce the incidence of hyperglycemia, dyslipidemia, hypertension, osteoporosis, and cardiovascular complications [57,58]. This finding is of crucial importance considering the correlation found between steroid use and NODAT development. A wider use of steroid-free immunosuppressive protocols is warranted. Post-transplant steroid-induced glucose homeostasis involves multiple mechanisms. Glucocorticoids promote adiposity and lipolysis, with increased release into the bloodstream of free fatty acids. They also inhibit protein synthesis, thus leading to enhanced protein breakdown in skeletal muscles. The latter event determines increased production of aromatic and branched-chain amino acids, which are ultimately associated with insulin resistance. Post-receptor insulin signaling dysfunction, another deleterious effect of chronic steroid administration, may favor liver steatosis directly or through inhibition of osteocalcin activity. Furthermore, glucocorticoids may reduce pancreatic β-cells’ survival, eventually leading to impaired insulin secretion [9]. Due to the small number of patients receiving cyclosporine, sirolimus, or everolimus included in our analysis, we could not detect any significant association between specific immunosuppressive drugs or schemes and NODAT. Nevertheless, both CNI [59,60] and mTORi [61,62] exhibit diabetogenic properties, dramatically increased by the concomitant use of steroids. 

Finally, comparing several clinical and laboratory parameters recorded during the first post-transplant year, we found that patients with NODAT exhibited worse renal function, higher systolic and diastolic blood pressure, increased total cholesterol and triglycerides levels, and higher uric acid concentrations than recipients with normal glucose metabolism. The association between diabetes (both pre- and post-transplant) with hypertension and lipid disorders is well-documented [48,55]. The causative relationship between NODAT and impaired renal function after KT is more difficult to define and should be better evaluated with properly designed prospective observational studies and serial histologic assessment of the allograft [9].

We recognize that our study has several limitations including retrospective nature, relatively small sample size, short-term follow-up, and lack of protocol allograft biopsies. Also, we could not assess the impact of different immunosuppressive regiments on NODAT susceptibility. However, it describes a homogeneous population of Italian KT recipients on a triple-agent CNI-based immunosuppressive scheme, consistently and rigorously managed by a dedicated multidisciplinary team. 

## 5. Conclusions

The prevalence of NODAT in our cohort of KT recipients was relatively high. Age at transplant, BMI, and total steroid dose within the first post-transplant month were associated with NODAT development. The systematic application of pre- and post-transplant screening protocols with OGTT, as well as tailored immunosuppression and prompt dietician referral could reduce NODAT-related complications. In the near future, studies addressing causal relationships, long-term patient- and allograft-related outcomes, and pre-emptive strategies are warranted.

## Figures and Tables

**Table 1 medicina-58-01608-t001:** Baseline characteristics of kidney transplant recipients with normal glucose metabolism (NGM), abnormal glucose metabolism (AGM), or new-onset diabetes mellitus after transplantation (NODAT).

	Overall (n = 522)	NGM (n = 386)	AGM (n = 56)	NODAT (n = 66)	ODM (n = 80)	
Variables	Means (± SD) or n (%)					*p*
Caucasian ethnicity	430/522 (82.4)	311/386 (80.6)	47/56 (83.9)	50/66 (75.8)	72/80 (90)	# 0.5494 † 0.3675 * 0.0448
Male	295/522 (56.5)	210/386 (54.4)	36/56 (64.3)	38/66 (57.6)	49/80 (61.25)	# 0.1642 † 0.6323 * 0.2620
Age at transplant (years)	48 ± 12	46 ± 12	51 ± 11	56 ± 9	56 ± 9	# 0.01 † <0.001 * <0.0001
BMI (kg/m^2^)	23 ± 3	23 ± 3	24 ± 3	25 ± 5	25 ± 4	# 0.08 † <0.0001 * <0.0001
Dialysis (HD/PD/None)	374/108/40	277/78/31	39/12/5	49/13/4	58/18/4	-
HD	374/522 (71.6)	277/386 (71.8)	39/56 (69.6)	49/66 (74.2)	58/80 (72.5)	# 0.7427 † 0.6778 * 0.8936
Dialysis vintage (months)	59 ± 54	59.3 ± 52.8	51.7 ± 32.3	63 ± 69	62.8 ± 74.3	# 0.30 † 0.51 * 0.62
Family history of diabetes	103/522 (19.7)	66/386 (17.1)	14/56 (25)	23/66 (34.8)	23/80 (28.75)	# 0.1512 † 0.0008 * 0.0158
Living donor	78/522 (14.9)	64/386 (16.6)	9/56 (16.1)	4/66 (6.1)	5/80 (6.25)	# 0.9236 † 0.0271 * 0.0178
Cold ischemia time (hours)	13.4 ± 4.0	13.3 ± 4.1	13.9 ± 3.5	13.3 ± 3.8	13.5 ± 4	# 0.24 † 0.84 * 0.77
Anti-HCV IgG +	35/522 (6.7)	25/386 (6.5)	3/56 (5.4)	8/66 (12.1)	7/80 (8.75)	# 0.7479 † 0.1033 * 0.4643
Anti-HBsAg IgG +	17/522 (3.3)	8/386 (2.1)	6/56 (10.7)	3/66 (4.5)	3/80 (3.75)	# 0.0005 † 0.2282 * 0.3684
Anti-CMV IgG +	435/522 (83.3)	315/386 (81.6)	47/56 (83.9)	58/66 (87.9)	73/80 (91.25)	# 0.6731 † 0.2149 * 0.0354

Note: Abbreviations: NGM, normal glucose metabolism; AGM, abnormal glucose metabolism; NODAT, new-onset diabetes after transplantation; ODM, overall diabetes mellitus; SD, standard deviation; HD, hemodialysis; PD, peritoneal dialysis; HCV, hepatitis C virus; HBV, hepatitis B virus; CMV, cytomegalovirus. #: NGM vs. AGM; †: NGM vs. NODAT; *: NGM vs. ODM.

**Table 2 medicina-58-01608-t002:** Primary renal diseases of kidney transplant recipients with normal glucose metabolism (NGM), abnormal glucose metabolism (AGM), or new-onset diabetes mellitus after transplantation (NODAT).

	Overall (n = 522)	NGM (n = 386)	AGM (n = 56)	NODAT (n = 66)	ODM (n = 80)	
Variables	n (%)					*p*
ADPKD	104/522 (19.9)	72/386 (18.6)	18/56 (32.1)	11/66 (16.7)	14/80 (17.5)	# 0.0191 † 0.7001 * 0.8088
Glomerular or immunologically mediated nephropathy	130/522 (24.9)	98/386 (25.4)	13/56 (23.2)	15/66 (22.7)	19/80 (23.7)	# 0.7258 † 0.6444 * 0.7583
Tubulo-interstitial or vascular nephropathy	143/522 (27.4)	95/386 (24.6)	21/56 (37.5)	19/66 (28.8)	27/80 (33.8)	# 0.0404 † 0.4702 * 0.0906
Other or unknown nephropathy	145/522 (27.8)	114/386 (29.5)	12/56 (21.4)	14/66 (21.2)	19/80 (23.7)	# 0.2092 † 0.1655 * 0.2971

Note: Abbreviations: NGM, normal glucose metabolism; AGM, abnormal glucose metabolism; NODAT, new-onset diabetes mellitus after transplantation; ODM, overall diabetes mellitus; ADPKD, autosomal dominant polycystic kidney disease; #: NGM vs. AGM; †: NGM vs. NODAT; *: NGM vs. ODM.

**Table 3 medicina-58-01608-t003:** Clinical and laboratory parameters at one-, six-, and 12-month follow-up of kidney transplant recipients with normal glucose metabolism (NGM) or new-onset diabetes mellitus after transplantation (NODAT).

	1 Month NGM	1 Month NODAT		6 Months NGM	6 Months NODAT		12 Months NGM	12 Months NODAT	
Variables	Mean (± SD) or Median (25–75°PCT)		*p*	Mean (± SD or Median (25–75°PCT)		*p*	Mean (± SD) or Median (25–75°PCT)		*p*
MDRD eGFR (mL/min)	57 ± 21	50.5 ± 23.4	0.03	56 ± 19	46.7 ± 15.3	0.001	57 ± 20	50.1 ± 16.5	0.02
SCr (mg/dL)	1.4 ± 0.5	1.5 ± 0.5	0.08	1.4 ± 0.5	1.5± 0.4	0.03	1.4	1.5 ± 0.4	0.04
Proteinuria (g/24 h)	0.12 (0.09–0.35)	0.25 (0.18–0.34)	0.3	0.19 (0.13–0.28)	0.20 (0.12–0.3)	0.09	0.16 (0.10–0.24)	0.22 (0.13–0.30)	0.45
BMI (kg/m^2^)	23 ± 4	25 ± 4	<0.0001	NA	NA	NA	24 ± 4	26 ± 4	0.006
SBP (mmHg)	131 ± 17	135 ± 19	0.21	131 ± 19	140 ± 25	0.001	130 ± 17	140 ± 21	<0.001
DBP (mmHg)	84 ± 10	81 ± 10	0.62	81 ± 10	82 ± 12	0.48	81 ± 10	81 ± 10	0.95
Uric Acid (mg/dL)	5.9 ± 1.5	6.2 ± 1.9	0.19	6.5 ± 1.5	7.1 ± 2.0	0.008	6.5 ± 1.5	7.3 ± 2.0	<0.001
Hemoglobin (g/dL)	11.1 ± 1.4	10.9 ± 1.3	0.34	12.3 ± 1.6	12.2 ± 1.7	0.76	12.8 ± 1.6	12.5 ± 1.7	0.24
Albumin (g/dL)	4.2 ± 0.4	4.1 ± 0.4	0.2	4.4 ± 0.3	4.4 ± 0.4	0.1	4.4 ± 0.4	4.5 ± 0.4	0.44
iPTH (pg/mL)	70.6 (40.5–115.1)	78.9 (5.4–142.8)	0.21	60.6 (38.2–96.7)	67.8 (33.8–175)	0.99	55.3 (37.6–94.0)	50.3 (32.7–139.9)	0.76
Calcium (mg/dL)	9.9 ± 0.8	9.7 ± 0.9	0.15	9.9 ± 0.8	9.9 ± 0.6	0.51	8.8 ± 0.7	9.8 ± 0.7	0.2
Phosphorus (mg/dL)	2.5 ± 0.9	2.5 ± 0.9	0.14	3.1 ± 0.7	3.2 ± 0.7	0.44	3.1 ± 0.7	3.2 ± 0.6	0.19
AP (U/l)	93 (70–126)	98 (70–131)	0.18	94 (69–128)	102 (79–134)	0.11	84 (62–112)	94 (72–123)	0.14
Glycemia (mg/dL)	80 ± 18	109 ± 32	<0.001	82 ± 12	110 ± 34	<0.001	79 ± 13	105 ± 32	<0.001
Insulinemia (ulU/mL)	8.4 (6.0–11.4)	9.7 (6.9–12.2)	0.78	7.6 (5.8–10.8)	8.7 (5.6–12.2)	0.27	8.6 (5.9–11.1)	8.1 (5.1–11.8)	0.49
HOMA index	2.5 (1.5–3.7)	2.4 (1.5–3.6)	<0.001	1.5 (1.1–2.2)	2.1 (1.4–3.1)	<0.001	1.6 (1.1–2.1)	1.9 (1.2–3.0)	0.003
HbA1c (mmol/mol)	35 ± 6	44 ± 8	<0.001	38 ± 7	47 ± 10	<0.001	38 ± 6	47 ± 11	<0.001
Total cholesterol (mg/dL)	217 ± 49	224 ± 59	0.28	204 ± 48	216 ± 56	0.07	199 ± 46	211 ± 51	0.05
HDL cholesterol (mg/dL)	62 ± 19	55 ± 25	0.02	57 ± 17	54 ± 24	0.49	58 ± 19	50 ± 14	0.003
Triglycerides (mg/dL)	163 ± 81	206 ± 148	<0.001	157 ± 112	189 ± 113	0.01	146 ± 67	173 ± 84	0.01
CRP (mg/dL)	0.3 (0.1–0.5)	0.4 (0.1–0.3)	0.19	0.1 (0.1–0.3)	0.2 (0.1–0.3)	0.01	0.1 (0.1–0.4)	0.3 (0.1–1.9)	0.6
25-OH-vitamin D (ng/dL)	13 ± 6	13 ± 8	0.66	14 ± 8	14 ± 9	0.96	16 ± 9	16 ± 12	0.88

Note: NGM, normal glucose metabolism; NODAT, new-onset diabetes after transplantation; SD, standard deviation; PCT, percentile; MDRD, modification of diet in renal disease; eGFR, estimated glomerular filtration rate; SCr, serum creatinine concentration; BMI, body mass index; SBP, systolic blood pressure; DBP, diastolic blood pressure; iPTH, intact parathyroid hormone; AP, alkaline phosphatase; HOMA, homeostasis model assessment; HDL, high-density lipoprotein; CRP, C-reactive protein.

**Table 4 medicina-58-01608-t004:** Multivariate analysis for NODAT development during the first year after kidney transplantation.

	B	SE	Wald	*p*	Exp (B)
Age at transplant	0.054	0.015	13.305	<0.001	1.056
Body mass index at T1	0.100	0.042	5.775	0.016	1.106
Serum creatinine concentration at T1	−0.040	0.247	0.026	0.872	0.961
Total steroid dose at T1	0.001	0.000	5.918	0.015	1.001
Constant	−7.755	1.261	37.843	<0.001	0.000

## Data Availability

The database is available for consultation upon formal request for publication purposes.

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
