# Peer review of "Prevalence and Risk Factors of Abnormal Glucose Metabolism and New-Onset Diabetes Mellitus after Kidney Transplantation: A Single-Center Retrospective Observational Cohort Study"

_medicina, 2022, doi:10.3390/medicina58111608_

Round 1

Reviewer 1 Report (Previous Reviewer 2)

Good manuscript.

Well organized

no remarkes to point. 

Reviewer 2 Report (Previous Reviewer 1)

Thanks for the opportunity to review this work. The authors have addressed my initial review concerns. I believe the manuscript merits publication. 

This manuscript is a resubmission of an earlier submission. The following is a list of the peer review reports and author responses from that submission.

Round 1

Reviewer 1 Report

This was a large single center retrospective analysis on NODAT prevalence and causation post kidney transplantation. 

Main suggestions:

1. At the baseline characteristics I could not find BMI even thought there is some mentioning downstream about the average BMI of the transplanted population.

2. Prediabetes is a common culprit for NODAT. Is there enough data to have this cofactored to the analysis? If not, perhaps it should be mentioned on the study limitations.

3. Impaired fasting glucose vs. impaired glucose tolerance. Perhaps IFG could be obviated?

4. Table 2. Not clear as to the relevance 

5. Table 5. A few points. How was the comparison performed? I would suggest refashioning the table perhaps eliminating some columns and calculating the RR for NODAT development for tac vs cya vs mTOR or even CNIs vs non-CNI. On a cursory check, RR for tac was 0.11, for cya was 0.17, and for mTOR was 0.06..

6. Table 7 ( multivariate analysis). What variables had been included in the analysis and how were they preselected? SCreat does not seem to be significant risk factor, contrary to what is said in the main text

Minor suggestions:

Ln 18. Change punctuation (".Were analyzed")

Ln 21. Define the abbreviation "T6"

Ln 214. Define abbreviation HOMA index

Ln 256. Please define RRT

Reviewer 2 Report

This paper explores an attractive thematic to the endocrine/nephrology community interested in diabetes.

The quality of language is very good and clear.

Overall, the paper is very well written.

Therefore, I have only a few remarks to the authors:

Abstract section

Methods. Data from 522 KT performed between 17 January 2004 and December 2014. Were analyzed.” – transform in 1 sentence.

Oral glucose tolerance test 20 (OGTT) was performed at T6 in 303 subjects. – please define T6  in the abstract– 6 months?

“Conclusions: The prevalence of NODAT in Italy 30 is relatively high. “ –  this single-center retrospective observational cohort study so you ca not generalize the conclusions to Italian population.

Introduction  section

You could also list the modifiable risk factors for NODAT.  Besides, sedentary behavior is a modifiable risk factor.

Reviewer 3 Report

The authors report the prevalence of glucose metabolism abnormalities in a cohort of Italian contemporary KT recipients, focusing on the early post-transplant course and investigating post-transplant modifiable risk factors.

This is a valuable study in these group of patients but there are some shortcomings:

- Patients with pre-transplant diabetes, multiple organ transplantation and retransplantation  were excluded? 

- is the any data about DGF (delayed graft failure)?

- The table 2 presents data from 303 patients and 219, while in the text, in lines, 172-176, the number of patients who did not take the test adds up to 214 (non-compliant patients (166/522, 31.8%) as well as those with pre-transplant history of diabetes mellitus (14/522, 2.7%) or NODAT diagnosis before T6 (34/522, 6.5%) were exempt from the test. Review the data.

- There are many tables. I suggest removing table 3 and presenting the data in the form of text and get together tables 1 and 4.

- Improve the discussion.

- Review the conclusion.